# Off-Policy Evaluation with Policy-Dependent Optimization Response

**Wenshuo Guo**[*]
Department of EECS
University of California, Berkeley
wguo@cs.berkeley.edu

**Michael I. Jordan**[*]
Department of EECS and Department of Statistics
University of California, Berkeley
jordan@cs.berkeley.edu

**Angela Zhou**[*]
Department of Data Sciences and Operations
University of Southern California
zhoua@usc.edu

## Abstract

The intersection of causal inference and machine learning for decision-making is rapidly expanding, but the default decision criterion remains an *average* of individual causal outcomes across a population. In practice, various operational restrictions ensure that a decision-maker's utility is not realized as an *average* but rather as an *output* of a downstream decision-making problem (such as matching, assignment, network flow, minimizing predictive risk). In this work, we develop a new framework for off-policy evaluation with *policy-dependent* linear optimization responses: causal outcomes introduce stochasticity in objective function coefficients. Under this framework, a decision-maker's utility depends on the policy-dependent optimization, which introduces a fundamental challenge of *optimization* bias even for the case of policy evaluation. We construct unbiased estimators for the policy-dependent estimand by a perturbation method, and discuss asymptotic variance properties for a set of adjusted plug-in estimators. Lastly, attaining unbiased policy evaluation allows for policy optimization: we provide a general algorithm for optimizing causal interventions. We corroborate our theoretical results with numerical simulations.

## 1   Introduction

The interface of causal inference and machine learning offers to "deliver the right intervention, at the right time, to the right person". An extensive line of research studies off-policy evaluation (OPE) and learning—evaluating the average causal outcomes under alternative personalized treatment assignment policies that differ from the treatment assignment which generated the data (and may have introduced confounding), so that one may optimize over the best such treatment rule [Manski, 2004, Dudík et al., 2011, Zhao et al., 2012, Thomas et al., 2015, Athey et al., 2017, Kitagawa and Tetenov, 2018, Kallus and Zhou, 2018b]. Most of this work is based on the assumption that the appropriate decision criterion is an *average* of individuals across a population. But various operational restrictions or settings imply that a decision-maker's utility is often not realized as an *average* but rather as an *output* of a downstream planning or decision-making problem.

For example, in studying the effects of price incentives in a matching market (e.g., on a rideshare platform), a firm's revenue is not realized until it matches riders to drivers under certain

---

[*]Authors listed in alphabetical order.

36th Conference on Neural Information Processing Systems (NeurIPS 2022).

constraints [Mejia and Parker, 2021, Ma et al., 2021]. While the marketplace may offer incentives to drive or accept rides and induce causal effects on individuals, the final utility is determined by the *new* matches, taking into account operational constraints and structure.

As another example, although job training (and personalized provision thereof) is commonly touted in causal inference and machine learning papers as a promising example for personalized treatment policy assignment [Athey et al., 2017, Kitagawa and Tetenov, 2018, Knaus et al., 2020], labor economists voice a general concern that "the possible existence of equilibrium effects on the efficiency of the programs seems quite real" [Crépon and Van Den Berg, 2016, p.541]. The equilibrium concern is that personalized provision of job training may not lead to actual beneficial gains at *the population* level due to externalities (substitution effects/congestion in matching) of the labor search process in a finite market. While impressive cluster-randomized trials have been deployed to assess these effects [Crépon et al., 2013], it would be useful if there exists a framework that can model the equilibrium effect and evaluate treatment policies directly based on available data of individual-level causal effects. In some settings, population-level impacts may be well-modeled as a downstream optimization response. The development of such a framework is our focus in the current paper.

We study a new framework for policy evaluation and optimization where there is a *personalized treatment policy* on individual-level outcomes, and a *policy-dependent* optimization response. The key difference between this model and previous work on off-policy evaluation and optimization is that: although treatments realize causal effects on *individuals*, a treatment policy's value depends on a further downstream *policy-dependent* optimization. We study how to evaluate different policies without bias (off-policy evaluation) and how to optimize for the optimal policy under this framework (policy optimization).

Our contributions are as follows: we first introduce the model of policy-dependent optimization response[2], which we formulate as a nonconvex stochastic optimization problem. For off-policy evaluation, we develop a framework of *policy-dependent optimization response*, decompose the bias that arises in this framework ("optimization bias") and show how to control it via the design of estimators for the policy-dependent estimand. Finally, we provide a general algorithm for optimizing causal interventions. We corroborate the theoretical results with experimental comparisons.

## 1.1 Related work

We highlight the most relevant work from causal inference, off-policy evaluation, and optimization under uncertainty in the main text. We include additional or tangential discussion in Appendix A.

There is an extensive literature on off-policy evaluation and optimization [see, e.g., Manski, 2004, Dudík et al., 2011, Zhao et al., 2012, Swaminathan and Joachims, 2015]. Relative to this line of work, we focus on the introduction of a downstream decision response, arising for example from operational constraints.

The case of constrained policies has been considered in the OPE literature. Our setting is conceptually different but overlaps in some application contexts. Specifically, we decouple the downstream *policy-dependent response*, i.e. over a similar constraint space, from treatment decisions that have causal effects. For example, Bhattacharya [2009] studies the setting of "roommate assignment" with discrete types; i.e., perfect bipartite matching. A crucial difference is that in their setting, the causal treatment of interest *is* the assignment decision to the other individual type; while in our setting the causal treatment only affects certain *parameters* of the assignment decision, such as edge costs. We instantiate an analogous example in our framework to highlight our decoupled causal intervention and prediction decisions. Consider a setting with a causal intervention, such as a diversity information intervention affecting a student's probability of getting along with various types. Here the policy performs treatments on individual's diversity information, and the final assignment decision is policy-dependent response. Other work considers resource-budgeted allocation Kube et al. [2019], which is structured because of reformulation of thresholds Lopez et al. [2020]. Sun [2021] studies sharp asymptotics for the additional challenge of stochasticity in the budget.

---

[2]For terminology, we use policy-dependent (optimization) response or downstream policy-dependent response to refer to the same concept.

Some works that illustrate the embedding of causal effect estimates in optimization-based decision problems include Rahmattalabi et al. [2022], although their formulation is ultimately a mixed-integer optimization.

In contrast to an extensive line of work on heterogeneous causal effect estimation [Shalit et al., 2017, Wager and Athey, 2018, Künzel et al., 2019], often crucially leveraging simpler structure of the treatment contrast rather than the conditional outcomes, in this work we require estimation of the latter due to the downstream optimization and distributional convergence for the perturbation method. In turn, combining causal outcome estimation with adjustments for optimization bias requires different properties of the estimation strategy, namely plug-in estimation of a modified regression model; we focus on estimators that modify the first-order conditions of a regression model to algebraically achieve an AIPW-type adjustment as discussed in Bang and Robins [2005]. See also Scharfstein et al. [1999], Tran et al. [2019], Shi et al. [2019], Chernozhukov et al. [2021].

This work focuses on the challenge of *optimization bias* for policy evaluation introduced in our setting, for *generic* linear optimization problems. This well-known challenge of in-sample optimization bias ("sample average approximation bias") fundamentally demarcates the statistical regime of optimization under uncertainty from sample mean estimation [Bayraksan and Morton, 2006, Shapiro et al., 2021]. Recent work develops bagging, jackknife, perturbation and variance-corrected perturbation approaches for bias adjustment [Lam and Qian, 2018, Ito et al., 2018, Kannan et al., 2020, Gupta et al., 2021]. We extend a perturbation method of Ito et al. [2018] to the setting of nonlinear predictions.

## 2 Preliminaries

We first define the setting for off-policy evaluation with policy-dependent responses. We distinguish between the *causal decision policy* $\pi$ and the *downstream optimization response* $x$. The causal decision policy $\pi$ intervenes on individual units, while the policy-dependent responses are solutions to a downstream optimization problem on the causal responses of all the units.

**1. Off-policy evaluation.** We first describe the single time step off-policy policy evaluation and optimization problem [see Dudík et al., 2014, Hirano and Porter, 2020, for further context]. Let covariates be $W \in \mathcal{W} \subseteq \mathbb{R}^d$, binary treatment be $T \in \{0,1\}$[3], and potential outcomes be $c(T)$. Denote the covariates' distribution as $\mathcal{P}$. Without loss of generality we consider lower is better for $c$; e.g. we minimize costs. We consider a setting of learning causal responses from a dataset of tuples $\mathcal{D}_1 = \{(W_i, T_i, c_i)\}_{i=1}^n$ where treatment is assigned randomly or in an observational setting; henceforth we call this the *observational / experimental dataset*.

We let $\pi_t \colon \mathcal{W} \mapsto [0,1]$ denote a personalized policy mapping from covariates to a (probability of) treatment $t$. Later we will focus on parameterized policies, such as $\pi_t(w) = sigmoid(\varphi^\top w)$ or policies that admit global enumeration. The goal of off-policy learning is to optimize the causal interventions (aka policies) by estimating average outcomes induced by any given policy. Throughout we will follow the convention that, a random variable $c(\pi_t)$ denotes $c(\pi_t) = c(t)\mathbb{I}(Z_t = t)$, where $Z_\pi \in \{0,1\}$ is a Bernoulli random variable of policy assignment: $Z_\pi \sim \text{Bern}(\pi_1)$. Then, the (random) outcome for a given covariate with policy $\pi$ is:

$$c(\pi) = \sum_t c(\pi_t) = \sum_t c(t)\mathbb{I}(Z_\pi = t).$$

The average treatment effect (ATE) of a policy $\pi(\cdot)$ is then $\mathbb{E}[c(\pi)]$, where the expectation is taken over the randomness of the covariates $W \sim \mathcal{P}$, assignments induced by $\pi$, and $c$ conditional on realized treatment $t$ and covariates.

**2. Policy-dependent responses.** *Policy-dependent optimization* solves a downstream stochastic linear optimization problem over a decision problem $x \in \mathcal{X} \subseteq \mathbb{R}^m$ on the $m$ units given a causal intervention policy. In particular, $m$ represents the dimension of the downstream decision problem. Relative to the downstream decision problem, causal outcomes may enter *either* as uncertain objective coefficients (in $c$) *or* constraint capacities (in $b$).[4]

---

[3]The extension to non-binary treatments is immediate.

[4]Throughout the text we focus on uncertainty in $c$ for notational clarity; strong duality implies the same results hold for uncertainty in the constraint right-hand-side, $b$. The decision is made conditionally on context information $W$ but prior to realizations of potential outcomes, aka a policy-dependent response.

**Dimensionality of the responses.** We consider two different asymptotic regimes: an *out-of-sample, fixed-dimension, fixed-$m$* regime and an *in-sample, growing-dimension, growing-$n$* regime. We formalize the former regime, the main focus of the paper, in the following assumption.

**Assumption 2.1** (Out-of-sample, fixed-dimension regime). As $n \to \infty$, the dimension of the optimization problem $m$, given by a new draw of contexts $\mathcal{D}_2 = \{W_i\}_{1:m}$ remains finite. The decision-dependent response on $m$ units is measurable with respect to $\mathcal{D}_2$. Let $c_i(\pi) \overset{\text{def}}{=} \mathbb{E}[c(\pi(w)|w = W_i]$, we have that the policy value $v_\pi^*$ is:

$$v_\pi^* = \mathbb{E}[\min_x \{\textstyle\sum_{i=1}^m c_i(\pi)x_i \colon Ax \le b\}]. \tag{1}$$

Assumption 2.1 defines our *policy-dependent estimand* in this regime. The expectation is taken over the randomness of the policy $\pi$ and the randomness of the finite samples $\{w_i\}_{i=1}^m$. The main text focuses on statements in the regime of Assumption 2.1. Evaluating regret with respect to a fixed dimension is standard or implicit in the predictive optimization literature.[5]

**Assumption 2.2** (In-sample, growing-dimension regime). As $n \to \infty$, the limit of the objective function is an expectation over contexts.[6] The estimand is:

$$v_\pi^* = \mathbb{E}[\min_x \{\mathbb{E}[c(\pi)x] \colon Ax \le b\}]. \tag{2}$$

Recall that a policy maps from covariates to a (probability of) treatment. Assumption 2.2 precisely takes an expectation over the two sources of randomness: the outer expectation is taken over the randomness of the policy, and the inner expectation is taken over the randomness of the covariates $w$.

The limiting object in the growing-dimension regime is a "fluid limit" or asymptotic regime: informally we assume a meaningfully constrained optimization in the limit. We instantiate our framework in the following example.

**Example 2.3** (Min-cost bipartite matching). Our framework is precisely motivated by the practical challenges in causal inference tasks, where the problem of "policy dependent" optimizations pops up repeatedly. For instance, for price incentives in a matching market (such as a rideshare platform), the revenue/welfare outcome is not realized until the riders and drivers are matched under constraints. As another example, consider a manager wants to assign agents to different jobs, and assigning an agent to a job is associated with some cost. Our goal is to assign each agent to at most one job such that the overall cost is minimized. To incentivize the workers to complete the jobs, the company might want to provide some bonus to the agents. However, the overall efficiency and total payments are not realized until all the assignments are determined.

The above type of application can be modeled as a min-cost bipartite matching problem, which is well known to have a totally unimodular linear relaxation. Clearly, the agents (or riders) and the jobs (or passenger requests) form the two sides of nodes for the matching. The edge costs in the matching stand for the cost or payment for an agent to complete that job. A treatment ($T = 1$) serves as intervention on the edge costs for that agent, and the covariates $W$ could be any observable features of the agents, such as preferences, demographic information, etc. Given any allocation rule of the bonuses, the manager faces a downstream min-cost bipartite matching:

$$\min_{x \in \{0,1\}^{|\mathcal{E}|}} \left\{ \textstyle\sum_{e \in \mathcal{E}} c_e(\pi)x_e \colon \sum_{e \in \mathcal{N}(i)} x_e = 1, \forall i \in \mathcal{V} \right\}. \tag{3}$$

Here $\mathcal{N}(i)$ is the set of all edges contains node $i$, the $c_e$ are the edge costs, and $x = \{x_e\}_{e \in \mathcal{E}}$ represents the matching where $x_e = 1$ means that edge $e$ is selected[7].

In Appendix E we include an additional example of predictive risk optimization, beyond linear optimization, which requires a different estimation strategy.

---

[5]The predictive optimization literature instead views each dimension of the decision variable as a multivariate outcome; relative to that, our regime can be interpreted as the setting of a scalar-valued contextual response.

[6]Assume the constraint $b$ scales with $n$ in a meaningful problem-dependent way so that constraints are neither all slack nor infeasible in the limit.

[7]In the later analysis we use the linear relaxation with $x_e \in [0, 1]$ (continuous interval). For bipartite matching because of *total unimodularity* the linear relaxation is tight and equivalent to integral formulation.

| | Out of sample, fixed $m$ (Assumption 2.1) | | In-sample, growing $n$ (Assumption 2.2, Appendix D) | |
|---|---|---|---|---|
| | Evaluation | Policy optimization | Evaluation | Policy optimization |
| **AIPW** | N/A | | Sample splitting (finite VC-dim $x$) | Uniform generalization requires problem-dependent structure (finite VC-dim $x$) |
| **WDM** | Perturbation method | Uniform generalization from out-of-sample risk bounds | Perturbation | |
| **GRDR** | Perturbation method | | Perturbation Doubly-robust estimation | |

Table 1: Summary of regimes and estimation properties. The main text provides methods for Assumption 2.1. Additional structural restrictions permit extensions for Assumption 2.2.

**3. Policy optimization with policy-dependent responses.** Putting together the pieces of the previous subsections, the off-policy optimization over candidate policies $\pi \in \Pi$ is:

$$\min_{\pi \in \Pi} \min_{x \in \mathcal{X}} \left\{ \sum_{i=1}^{m} c_i(\pi) x_i \colon Ax \le b \right\}, \tag{4}$$

where $m$ represents the dimension of the decision problem (e.g., the number of edges in Example 2.3), and $x$ denotes the whole response vector $\{x_i\}_{i \in [m]}$.

We illustrate this framework by revisiting our examples.

**Example 2.4** (Policy optimization for Example 2.3, min-cost matching). In the min-cost bipartite matching example, the optimal assignments with a given policy $\pi$ can be solved via the linear program in Equation (3). Suppose that we want to find the best intervention policy which gives the lowest matching cost. Then, the policy optimization problem is:

$$\min_{\pi \in \Pi} \min_{x \in \mathcal{X}} \left\{ \sum_{e \in \mathcal{E}} c_e(\pi) x_e \colon \sum_{e \in \mathcal{N}(i)} x_e = 1, \forall i; x_e \ge 0, \forall e \right\},$$

where $\Pi$ denotes the set of all policies that are of interest.

# 3 Problem Description: Optimization Bias

We focus on off-policy evaluation in view of the downstream optimization over the decision variables $x = \{x_i\}_{i \in [m]}$. We first discuss *plug-in* estimation approaches without causal adjustment to introduce the challenge of optimization bias in this regime. We then discuss causal estimation in Section 4.

**From estimation bias to optimization bias.** Denote $\mu_t(w) = \mathbb{E}[c(t) \mid W = w]$ as the conditional outcome mean of the population with treatment $t$ and covariates $w$. We consider "predict-then-optimize" approaches which learn some $\hat{\mu}_t(w) = \mathbb{E}[c \mid W = w, T = t]$ and optimize with respect to it, so that our estimator is:

$$\hat{v}_\pi = \min_{x \in \mathcal{X}} \left\{ \sum_{i=1}^{m} \sum_{t \in \{0,1\}} \pi_t(w_i) \hat{\mu}_t(w_i) x_i \colon Ax \le b \right\}.$$

Note that due to the estimation and minimization step, $\hat{v}_\pi$ is not an unbiased estimator for $v_\pi^*$. Define the overall error of $\hat{v}_\pi$ with respect to the target estimand of Equation (1) as: $\text{err} = v_\pi^* - \mathbb{E}[\hat{v}_\pi]$. We decompose the overall error into two parts: the estimation bias of the plug-in estimator, and the optimization bias. Denote $\tilde{v}_\pi$, the best-in-class feasible estimate using the true conditional expectations $\mu_t^*$:

$$\tilde{v}_\pi = \min_{x \in \mathcal{X}} \left\{ \sum_{i=1}^{m} \sum_{t \in \{0,1\}} \pi_t(w_i) \mu_t(w_i) x_i \colon Ax \le b \right\}.$$

Then, the estimation and optimization biases are: (by triangle inequality, $|\text{err}| \le |\text{bias}_\text{est}| + |\text{bias}_\text{opt}|$)

$$\text{bias}_\text{est} = \mathbb{E}[\hat{v}_\pi] - \mathbb{E}[\tilde{v}_\pi], \qquad \text{bias}_\text{opt} = v_\pi^* - \mathbb{E}[\tilde{v}_\pi].$$

**In-sample estimation bias due to optimization.** It is well known that in-sample estimation of the value of optimization problems is biased; e.g., $\hat{v}$ is a biased estimate for the true objective value $v_\pi^*$ due to optimization. Ito et al. [2018] studies a bias correction for affine linear objectives with an unbiased estimate of a parameter $\theta$. To understand the source of the bias due to optimization, observe that clearly $\sum_{i=1}^{m} \mu_t(w_i) x_i \ge \min_x \sum_{i=1}^{m} \mu_t(w_i) x_i$. The inequality remains valid when evaluating expectations over training datasets so that $\mathbb{E}[\sum_{i=1}^{m} \mu_t(w_i) x_i] \ge \mathbb{E}[\min_x \sum_{i=1}^{m} \mu_t(w_i) x_i]$. Noting that the RHS is the true objective $v_\pi^*$, we obtain in general the well-known optimistic bias, that $\mathbb{E}[\tilde{v}_\pi] \ge v_\pi^*$. In the policy evaluation setting, our estimates converge to the LHS, $\tilde{v}_\pi$, so that our estimator $\hat{v}_\pi$ is in general a *biased* estimate of the decision-dependent policy value even if we obtain *unbiased* estimates of the cost coefficient.

# 4 Causal Estimation with Policy-Dependent Responses

In this section we present an estimation approach building upon a perturbation method that adjusts for the aforementioned optimization bias. We summarize tradeoffs among estimation strategies in different regimes in Table 1 and possible extensions and additional structure in Appendix D.

## 4.1 Estimating causal effects: estimation bias

**Assumption 4.1** (Ignorability, overlap, SUTVA). For all $t$, $c(t) \perp\!\!\!\perp T \mid W$. The evaluation policy is absolutely continuous with respect to treatment probabilities in the training dataset. Assume the stable unit treatment value assumption.

**Confounding-adjusted plug-in estimators.** In general, plug-in estimation of $\hat{\mu}_t(W)$ does *not* admit unbiased predictions because of selection bias and model misspecification. Existing importance-sampling based estimators, e.g. the inverse propensity weighting (IPW) estimator and the doubly-robust augmented inverse probability weighting (AIPW) uses the propensity score to adjust confounding, under Assumption 4.1. Note importance sampling cannot *directly* be applied in our main regime of interest with out-of-sample evaluation as in Assumption 2.1, see Appendix B for a detailed overview.

We depart from previous work in off-policy evaluation, in view of the optimization bias adjustment (detailed in the next section), and study estimation methods that are *plug-in estimates* for OPE: $\mathbb{E}[c(\pi)] = \sum_t \mathbb{E}[\pi_t(W)\hat{\mu}_t(W)]$, for some outcome model $\hat{\mu}_t$ that is confounding-adjusted.

Note that IPW/AIPW-type estimators cannot be applied in the out-of-sample regime of Assumption 2.1. However, we may obtain out-of-sample risk bounds on the decision regret in this regime by virtue of out-of-sample generalization risk bounds on the generated regressors. We include more detailed discussion in Appendix D.3.

**Weighted direct method (WDM).** Outcome regression, learning $\hat{\mu}_t(W) = \mathbb{E}[c \mid T = t, W]$ directly from $\mathcal{D}_1$, is sometimes called the *direct method*. However, when $\hat{\mu}$ is a misspecified regression model such a method incurs bias. Nonetheless, re-weighting the estimation $\hat{\mu}$ (maximum likelihood, empirical risk minimization) by the inverse probability weights $1/e$ is known to adjust for the covariate shift; by a similar argument as that of Shimodaira [2000], Cao et al. [2009], Wang et al. [2019]. We call this approach *weighted direct method* (WDM), which solves:

$$\hat{\mu}_t^{\text{WDM}} \in \arg\min_\mu \mathbb{E}\left[\frac{\mathbb{I}(T=t)}{e_t(W)}(c - \mu_t(W))^2\right]. \tag{5}$$

**Doubly-robust direct method (GRDR).** We also consider an approach that achieves doubly-robust estimation of the treatment-effect due to Bang and Robins [2005]. [See also Scharfstein et al., 1999, Tran et al., 2019]. This approach has been used for CATE estimation Shi et al. [2019], Chernozhukov et al. [2021]. The inverse propensity score reweighted treatment indicator is added as a covariate in the model, inducing coefficients $\epsilon_0, \epsilon_1$. Define

$$\hat{\mu}^{\text{GRDR}} = \mu(W) + \epsilon_1(T/e_1(W)) + \epsilon_0((1 - T)/e_0(W)).$$

Optimizing over $\hat{\mu}$ by (nonlinear) least-squares yields the following first-order optimality conditions for $\theta^{\text{GRDR}} = [\bar{\theta}, \epsilon_1, \epsilon_0]$:

$$\mathbb{E}[(c - \hat{\mu})\nabla_\theta\hat{\mu}] = 0, \mathbb{E}[(c - \hat{\mu})(T/e_1(W))] = 0, \mathbb{E}[(c - \hat{\mu})((1 - T)/e_0(W))] = 0. \tag{6}$$

Bang and Robins [2005] show that the first-order optimality conditions ensure that plug-in estimation of an average treatment effect with the model is equivalent to AIPW, hence doubly-robust. Because it is designed primarily for estimation of the ATE, its use as an outcome predictor is more speculative. Although one can verify that its output is covariate-conditionally equivalent to CATE in expectation, and one can use this fact to again regress upon the pseudooutcomes, this final procedure would require re-verifying asymptotic convergence; we don't outline those arguments here. We include further discussion on the different estimation interpretations of GRDR in the two regimes in Appendix D.3.

## 4.2 Estimating the decision-dependent estimand

Our procedure is adapted from the perturbation method of Ito et al. [2018] which we describe here for completeness; we extend it from linear to nonlinear predictors. The method of Ito et al. [2018]

---

**Algorithm 1** Perturbation method, Alg. 2 of Ito et al. [2018])

---

**input** Estimation strategy $\diamond \in \{\text{WDM}, \text{GRDR}\}$; $h$: finite different parameter; $\pi$: policy.

0: Estimate $\hat{\xi}_\diamond = [\hat{\theta}_\diamond, \hat{\gamma}_\diamond]$ for $\hat{\mu}^\diamond$ from $\mathcal{D}_1$

1: $\hat{v}^{(0)} \leftarrow \min_{x \in \mathcal{X}} \sum_{i=1}^m x_i \sum_{t \in \{0,1\}} \pi_t(w_i) \hat{\mu}_t^\diamond(w_i; \hat{\xi}_\diamond)$

2: Generate $\{\xi_\diamond^{(j)}\}_{j=1}^s$: if by parametric bootstrap, learn $\hat{\xi}_\diamond^{(j)}$ from $\frac{N}{(1+h)^2}$ samples randomly chosen from $\mathcal{D}_1$ with replacement.

   Otherwise if using $\hat{\Sigma}$, estimator of asymptotic variance of $\xi$, approximate the distribution of $\xi^* + (1+h)\delta$. Add $\hat{\xi}$ to $\hat{\theta}$ where $\hat{\delta} \sim N(0, \frac{(1+h)^2-1}{N}\hat{\Sigma})$. Then set $\hat{\xi}_\diamond^{(j)} = \hat{\xi} + \hat{\delta}_j$.

3: **for** $j = 1, \ldots, S$ : **do**

4: $\quad \hat{v}^{(j)} \leftarrow \min_{x \in \mathcal{X}} \sum_{i=1}^m x_i \sum_{t \in \{0,1\}} \pi_t(w_i) \hat{\mu}_t^\diamond(w_i; \hat{\xi}_\diamond^{(j)})$.

5: **end for**

6: Output $\rho_h = \hat{v}_0 - \frac{1}{h}(\hat{v}^{(0)} - \frac{1}{s}\sum_{j=1}^s \hat{v}^{(j)})$.

---

focuses on one parameter that we denote $\xi = [\theta, \gamma]$, where we assume as outlined in Assumption 4.4 that it encompasses parameters of the outcome and propensity model (respectively). Define the policy-induced outcome model, $\mu_\pi(w) = \sum_t \pi_t(w)\mu_t(w)$, the estimation error $\delta = \hat{\xi} - \xi^*$, and the (parametrized) optimal solution at a given predictive model $x(\xi)$. The perturbation method is motivated by a finite-difference approximation to the optimization bias induced by estimation error $\delta$. Define the auxiliary functions given a scalar $\epsilon$ parametrizing the direction of $\delta$:

$$\eta(\epsilon) = \mathbb{E}_\delta \left[\sum_{i=1}^m x(\xi^* + \epsilon\delta)\pi(W; \xi^*)\right], \quad \phi(\epsilon) = \mathbb{E}_\delta \left[\sum_{i=1}^m x(\xi^* + \epsilon\delta)\pi(W; \xi^* + \epsilon\delta)\right].$$

We require regularity conditions for derivatives of these functions to exist:

**Assumption 4.2** (Perturbation method assumptions). (i) The optimal solution $x(\xi)$ is unique. (ii) $\hat{\xi}$ is an unbiased estimator of $\xi^*$.

We generalize Prop. 3 of Ito et al. [2018] for nonlinear models.

**Proposition 4.3.** *We have* $\eta(\epsilon) = \phi(\epsilon) - \epsilon\phi'(\epsilon) + O(\epsilon^2)$.

The plug-in estimated optimal value $\hat{v}_\pi$ unbiasedly estimates $\phi(1)$. Note $\phi'(1)$ is equivalent to the value of the bias. The perturbation method estimates $\phi'(1)$ by $(\phi(1+h) - \phi(1))/h$ for some small $h$.

It remains to estimate $\phi(1 + h)$. First we obtain $s$ samples of the perturbed parameter $\hat{\xi}_h = \xi^* + (1+h)\delta$, denoted as $\{\hat{\xi}_h^{(j)}\}_{j=1}^s$. Each replicate of $\hat{\xi}^{(j)}$ leads to an optimization estimate $\hat{v}^{(j)} = \min_x \sum_{i=1}^m x_i \hat{\mu}_\pi(w_i, \hat{\xi}_h^{(j)})$. The debiased estimator is:

$$\rho_h = \hat{v}^{(0)} - \frac{1}{h}(\hat{v}^{(0)} - \frac{1}{s}\sum_{j=1}^s \hat{v}^{(j)})$$

Our Proposition 4.3 then implies asymptotic unbiasedness (cf. Prop. 4 of Ito et al. [2018]) so that $\lim_{h \to 0} \mathbb{E}[\rho_h] = \mathbb{E}[\min_x \sum_{i=1}^m x_i \hat{\mu}_\pi(w_i, \xi^*)]$. We summarize the method in Algorithm 1.

**Asymptotic variance of estimation methods.** We discuss the asymptotic variance of the *weighted direct method* and GRDR via classical asymptotic analysis of *generated regressors* (specifically, stacked estimation equations of GMM) [Newey and McFadden, 1994]. We summarize this framework in Appendix B.1 for completeness and include the main result here that we invoke.[8]

**Assumption 4.4** (Estimators via GMM with generated regressors). Suppose the propensity score $e$ and outcome model $\mu$ are indexed by true parameters $\gamma^*, \theta^*$ that solve the respective estimating equations $\mathbb{E}[h(W, \gamma^*)] = 0$, $\mathbb{E}[g(W, \theta^*, \gamma^*)] = 0$. The functions $e_t(w), \mu_t(w)$ are in a Donsker class.

*Remark* 4.5 (Strength of assumptions). Algorithm 1 requires both unbiased and asymptotically normal predictions—stronger conditions than merely inference on the ATE. The Donsker assumption preserves asymptotic normality with generated regressors. The framework allows for nonparametric estimation via linear sieves (but not some high-dimensional regimes; see Ackerberg et al. [2012]).

---

[8]Asymptotic normality of these approaches is taken as given in Cao et al. [2009], Bang and Robins [2005] and so we include these statements for completeness. For exposition and context of Donsker-type conditions in semiparametric inference, see Kennedy [2016] or other references.

---

**Algorithm 2** Subgradient method for policy optimization

---

1: **Input:** step size $\eta$, linear objective function $f$.

2: **for** $j = 1, 2, \cdots$ **do**
3:     At $\varphi^k$, obtain a subgradient in subdifferential $\mathcal{S}^*(\pi_\varphi^k) = \{x^* : f(x^*; \pi_\varphi^k) = \min_x f(x; \pi_\varphi^k)\}$
4:     Compute subgradient $\nabla_\varphi(\min_x f(x; \pi_\varphi^k)) \leftarrow \nabla_\varphi f(x^*; \pi_\varphi)$
5:     Update subgradient step: $\varphi^{k+1} \leftarrow \varphi^k - \eta \nabla_\varphi \left( \min_x f(x; \pi_\varphi^k) \right)$
6: **end for**

---

**Theorem 4.6** (Thm. 6.1, eq. 6.12 of Newey and McFadden [1994])**.** *Suppose Assumption 4.4 holds. Let* $\hat{G}_\alpha, \hat{G}_\theta, \hat{H}$ *denote the Jacobian matrices of partial derivatives of the moment conditions* $g, h$ *with respect to the respective parameters, i.e.* $\hat{G}_\gamma = n^{-1} \sum_{i=1}^n \nabla_\gamma g(w_i, \hat\theta, \hat\gamma)$. *Let* $\hat{V}_\gamma = (\hat{H}^{-1} \hat{h}_i)(\hat{H}^{-1} \hat{h}_i)^\top$. *Then an estimator of the asymptotic variance is:*

$$\hat{V}_\theta = \hat{G}_\theta^{-1} \left( n^{-1} \sum_{i=1}^n \hat{g}_i \hat{g}_i^\top \right) (\hat{G}_\theta^{-1})^\top + \hat{G}_\theta^{-1} \hat{G}_\gamma \hat{V}_\gamma \hat{G}_\gamma^\top (\hat{G}_\theta^{-1})^\top.$$

Since $\hat{V}_\gamma$ depends only on the specification of the propensity score, to completely specify the asymptotic variance for the above formula we state the mixed terms $\hat{G}_\gamma, \hat{G}_\theta$.

**Proposition 4.7** (Asymptotic normality of WDM)**.** *Let* $e_t(w), \mu_t(w)$ *satisfy Assumption 4.4 with the moment condition* $g_t(W, \theta, \gamma) = e_t(W; \gamma)^{-1}(c - \mu_t(W; \theta))^2$ *and* $g = [g_0, g_1]$. *Then*

$$\hat{G}_\gamma = \begin{bmatrix} \mathbb{E}_n[2T(c - \mu(W; \theta)) \frac{\partial}{\partial \theta}(e_1^{-1}(W, \gamma)) \frac{\partial \mu}{\partial \theta}] \\ \mathbb{E}_n[2(1 - T)(c - \mu(W; \theta)) \frac{\partial}{\partial \theta}(e_0^{-1}(W, \gamma)) \frac{\partial \mu}{\partial \theta}] \end{bmatrix}.$$

These formulas are generally computable from standard output of optimization solvers for nonlinear least squares: gradients and Hessians. In practice, using the parametric bootstrap may be simpler at a higher computational cost.

## 4.3 Optimizing Causal Interventions

Algorithm 1 provides estimation for a fixed policy. We now discuss how to optimize over policies; e.g., implementing the outer optimization over policies $\min_{\pi \in \Pi}$ in Equation (4). We focus on the case where the policy $\pi_t(w)$ is parametrized by and differentiable in a parameter $\varphi \in \Psi$. For example, for the logistic policy parameterization, $\pi_t(w) = sigmoid(\varphi_t^\top w)$. We consider a robust subgradient method, based on Danskin's theorem, detailed in Algorithm 2. Such an approach is a common heuristic used in adversarial machine learning.

We solve the inner optimization problem to full optimality in line 3 and take (sub)gradient steps for the outer optimization. We evaluate (sub)gradients of the inner optimization solution in line 3 by evaluating the gradient of the objective with respect to $\varphi$, fixing the inner optimization variable $x^*$. Danskin's theorem implies that $\nabla_\varphi$ is a subgradient [Danskin, 1966]. The inner minimization can be solved via a linear optimization oracle for any fixed choice of policy. This use of the linear optimization oracle can be beneficial when special problem structures, such as matching and network flows, may also admit readily-available algorithmic solutions to full optimization.

The perturbation method is compatible with our optimization procedure because the bias-adjusted perturbation estimated from Algorithm 1 is affine in the optimization problems corresponding to each parameter replicate. Hence, run Algorithm 1 with an expanded linear objective over the $s$-product space $x' \in \mathcal{X}^s$ where $f(\tilde{x}, \pi) = \hat{v}_\pi^{(0)}(\tilde{x}_0) - \frac{1}{h}(\hat{v}_\pi^{(0)}(\tilde{x}_0) - \frac{1}{s} \sum_{j=1}^s \hat{v}_\pi^{(j)}(\tilde{x}_j))$.

So, re-optimize $\tilde{x}_j^* \in \arg\min_{x \in \mathcal{X}} \sum_{i=1}^m x_i \hat{\mu}_\pi^\diamond(w_i; \hat{\xi}_\diamond^{(j)})$ and apply Danskin's theorem to each optimization problem in the sum over $\hat{v}_\pi^{(j)}$ comprising $f(x', \pi)$. In fact, though adversarial machine learning focuses on min-max rather than our min-min optimization problem, this particular approach is simply subgradient descent on a nonconvex function (the solution to the inner optimization).

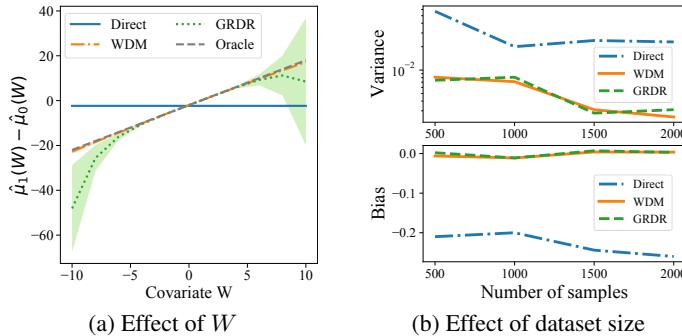

(a) Effect of $W$          (b) Effect of dataset size

Figure 1: *(In-sample estimation of $\hat{\mu}_1(W) - \hat{\mu}_0(W)$, with model mis-specification).* Comparison of direct / WDM / GRDR to the oracle. (a) Conditional estimation error averaged over ten random train sets; shaded area indicates std. error. (b) Bias / variance comparison with varying training data size.

## 5 Experimental Evaluation

Since real data suitable for both policy evaluation and downstream optimization is unavailable, we focus on synthetic data and downstream bipartite matching. We first illustrate estimation properties of the different approaches before showing the improvement obtained via policy optimization. Though we are not aware of prior approaches that are directly comparable for optimizing causal policy with a downstream optimization-dependent response, we include more comparisons to nonparametric estimators (e.g. causal forests [Wager and Athey, 2018]), and full implementation details. All code will be published.

**1. Causal effect estimation.** First, we investigate and illustrate the properties of different estimators. We generated dataset $\mathcal{D}_1 = \{(W, T, c)\}$ with covariate $W \sim \mathcal{N}(0, 1)$, confounded treatment $T$, and outcome $c$. Treatment is drawn with probability $\pi_t^b(W) = (1 + e^{-\varphi_1 W + \varphi_2})^{-1}$, $\varphi_1 = \varphi_2 = 0.5$. The true outcome model is given by a degree-2 polynomial,[9] $c_t(w) = poly_\theta(t, w) + \epsilon$, where $\epsilon \sim \mathcal{N}(0, 1)$. . In Figure 1a and 1b, we illustrate the (covariate-conditional) estimation error of the three estimators. In the mis-specified setting that induces confounding, the outcome model is a vanilla linear regression over $W$ without the polynomial expansion. The direct method results in more bias under mis-specification, while WDM and GRDR are robust as expected.

**2. Policy evaluation.** We compare the perturbation method (Algorithm 1) with three different estimators (direct, WDM, and GRDR). In both the well-specified / mis-specified model setting, we evaluate the mean-squared-error (MSE) of the estimated policy value with the three estimators, where the MSE is computed with regard to the ground-truth outcome model. Training data size $n$ increases from 500 to 2000 samples. We scaled the MSE down by the number of edges (a constant) and computed the MSE in terms of the averaged cost per edge in the matching.

For the policy-dependent optimization, we evaluate a min-cost bipartite matching problem, where the causal policy intervene on the edge costs (as detailed in Example 2.3). Specifically, the bipartite graph contains $m = 500$ left side nodes $W_1, \cdots, W_m$, and $m' = 300$ right side nodes. The policy $\pi_t$ applies treatments to the left side nodes and the outcome is the edge cost of edges with that node. While we grow the training data size, we fixed $m, m'$ (with $m > m'$) and evaluate over ten random draw of train/test data for each value of $n$. Figure 2 plots the results. When there is mis-specification, even a large training dataset cannot bring bias correction for the direct method, where both WDM and GRDR enjoy smaller and decreasing MSE.

We also conduct an ablation study for the corresponding performance in the mis-specified setting (i.e., no bootstrapping in Alg. 1). Results indicate that the perturbation method is helpful for MSE reduction for both WDM and GRDR. We further conduct evaluations with different bootstrap replicates' sizes, and the above conclusions remain robust for different replicate sizes (additional results in Appendix F).

**3. Policy optimization.** Lastly, we integrate the policy evaluation and the sub-gradient method (Alg 2) to conduct policy optimization. At each iteration of Alg 2, the perturbation algorithm (Alg 1) and one of the three different estimation methods are applied to evaluate the policy objective. We consider

---

[9]If not stated otherwise we spread the coefficients as $poly_\theta(t, w) = (1, w, t, w^2, wt, t^2) \cdot ([5, 1, -1, 2, 2, -1])^\top$. Additional supporting experiments under other nonlinear data-generating processes are in Appendix F.

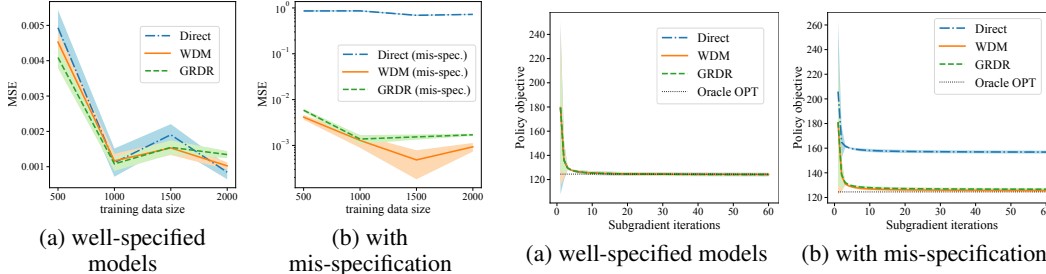

(a) well-specified        (b) with            (a) well-specified models   (b) with mis-specification

models        mis-specification

Figure 2: *(Policy evaluation via perturbation method (Algorithm 1).* Comparison of direct / WDM / GRDR estimators over increasing size of training data (averaged over ten runs).

Figure 3: *(Policy optimization).* Subgradient policy optimization with direct / WDM / GRDR estimation methods and a fixed test set. Averaged over ten random training datasets of size=1000.

a logistic policy $\pi_t(W) = sigmoid(\varphi_1 \cdot W + \varphi_2)$. To study the convergence and the effectiveness of the subgradient algorithm for minimization, we fix a test set and perform subgradient descent over 60 iterations for each run. We average the policy values at each iteration over ten runs, where in each run we generate a random set of training data and a random initialization of the starting policy.

We compare to the oracle estimator using the ground truth outcome model (Oracle OPT). Results are presented in Figure 3. Again, WDM and GRDR quickly converge to the oracle estimation, while the large bias of the direct method leads to poor policy optimization. We further evaluate the impact of average random selected initial policies to the performance, and compared Figure 3 with the results using a fixed initial policy. We observe that in this relatively low-dimensional example, the policy value converges to estimation-oracle-optimal after a few iterations (additional results and full training details in Appendix F).

# 6 Conclusion

We studied a new framework for causal policy optimization with a *policy-dependent* optimization response. We proposed evaluation algorithms and analysis to address the fundamental challenge of an additional optimization bias. Simulations for both the policy evaluation and optimization algorithms demonstrate the effectiveness of this approach. Interesting further directions include studying individual fairness of optimal allocations in applications such as school assignments or job matching, and/or computational improvements to the policy optimization algorithms.

# Acknowledgements

We wish to acknowledge support from the Vannevar Bush Faculty Fellowship program under grant number N00014-21-1-2941. WG acknowledges support from a Google PhD fellowship. AZ acknowledges support from the Foundations of Data Science Institute.

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
