# OpenReview forum: "Off-Policy Evaluation with Policy-Dependent Optimization Response"
_NeurIPS.cc/2022/Conference — NeurIPS 2022 Accept_

### Official Review · Reviewer_J5Rp · 2022-06-20

**Rating:** 4
**Confidence:** 3
**Soundness:** 2 fair
**Presentation:** 1 poor
**Contribution:** 2 fair

**Summary:**

This paper proposed an off-policy evaluation method through the output of a downstream decision making problem, which is interpreted as a policy-dependent optimization response. Optimization bias is addressed by extending an existing perturbation method for bias adjustment in parameter estimation to nonlinear predictions. The parameters of interest are of the outcome and propensity model. In the meantime, policy optimization is achieved by a subgradient method. Experiments conducted on simulated data show contribution from each algorithmic component. No comparisons to any baselines seem to have been shown.

**Questions:**

1 does it mean you can have a policy trained on one task whilst evaluated on another task?
2 how is Eq. 4 different from a multi-task RL objective
3 from line 177 the estimation bias and optimization bias seem to be two components of the overall error, whereas from line 181 the estimation bias becomes due to optimization. Is there not any contradiction?

**Limitations:**

Yes

**Strengths And Weaknesses:**

Main strengths:
1 Clear assumptions, algorithm details and experimental setup
2 addressed a meaningful problem of optimization bias in OPE

Main weaknesses:
1 no comparisons to appropriate previous work
2 some variables are left unexplained e.g. what is constraint b, and terminologies unaligned e.g. is causal intervention equivalent to policy?
3 unclear whether the various definitions are self-defined or extensively adopted since there are hardly any references made, and it's hard to tell which parts are the authors' own takeaway message

---

> ### Author Response · Authors · 2022-08-02
> **Response to reviewer J5Rp**
>
> Dear reviewer,
>
> Thank you for your thoughtful review! We hope our response answers your questions.
>
> # comparison to prior works
>
> In the experimental section, we compared the proposed method with a set of common benchmark estimators in the experimental section, including the direct method, WDM and GRDR. The direct method is the relevant baseline for the typical “estimate-then-optimize” approach without causal adjustment.
>
> We are not aware of prior approaches that are directly comparable for optimizing causal policy with a downstream optimization dependent response –this originality and novelty were also confirmed by other reviewers.
>
> Though there are not prior works, in Appendix E.4, we compared against popular CATE estimators:
> We compared the DM/WDM/GRDR to the Causal Random Forests estimator proposed in (Wager and Athey, 2018) following the setup in Section 5.1.
> We also compared how the CATE estimators affect the policy evaluation task with the perturbation method (section 5.2). Results show that the non-parametric random forest estimator is indeed unbiased, while the naive direct method has a large bias under model mis-specification.
>
> We will include these additional comparisons to the main text.
>
> # Is causal intervention equivalent to policy; what is constraint $b$
>
> As detailed in line 107, a policy $\pi$ maps from covariates to a (probability of) treatment, aka a causal intervention. In equation (1), matrix $A$ and vector $b$ define a linear constraint of feasible space for $x$; this is the standard form for linear optimization.
>
> # Contribution clarifications
>
> Our preliminary section (section 2.1) includes the causal effect estimation and off-policy evaluation framework, which is standard based on the literature. Section 2.2 explains the policy-dependent responses in our framework. Section 2.3 presents our main task, policy optimization with such policy-dependent responses.
>
> The main difficulty is such policy-dependent responses introduce an *optimization bias*, in line 180. Even existing estimators may achieve zero estimation bias (in line 180) in the large-sample limit, the optimization bias is still there. We propose methods to adjust for it in Section 4.
>
> Our main contribution lies in the novelty and technical contribution of analyzing the policy optimization problem with the policy-dependent responses. As confirmed in other reviews, such a problem is important in practical applications but has not been studied in previous off-policy evaluation works.
>
> We will add a paragraph in Section 1 to clarify the contribution.
>
> # about the optimization bias
>
> “..from line 181 the estimation bias becomes due to optimization”
>
> No contradiction. In line 181, the “in-sample estimation bias due to optimization” referred to the overall bias in the estimation of the target value $v_{pi}^\ast$ and was different from the term $bias_{est}$. To avoid any confusion we will revise the heading to “in-sample bias”. In the case of plug-in estimator approaches, bias in the optimization value can be further decomposed to $bias_{est}$ and $bias_{opt}$.
>
> # difference to a multi-task RL objective
>
> The setting is completely different from multi-task reinforcement learning: we study (single time-step) causal policy evaluation and optimization. Although Eq(4) might look  superficially similar to a multi-task RL objective, summing over $i$ sums over units that receive interventions (rather than, say, different objectives). There is a single objective function in our work subject to constraints, rather than constraints on multiple objectives (multiple outcomes. We defer to extensions and potential applications to multi-task RL for future work.
>
> #  Does it mean you can have a policy trained on one task whilst evaluated on another task?
>
> Our focus is different from training a policy on one task while evaluating on another. As detailed in Section 2 (in particular section 2.3), we aim to optimize a causal policy for which the objective contains an inner *policy-dependent optimization*.

---

> > ### Comment · Reviewer_J5Rp · 2022-08-02
> > **Response to Rebuttal**
> >
> > Thanks for your clarification. I'd like to increase my rating to borderline accept.

---

> > > ### Author Response · Authors · 2022-08-03
> > > **Author reply reviewer J5Rp**
> > >
> > > Dear reviewer,
> > >
> > > Thank you very much for the quick reply and we are glad that our response resolved some of the previous concerns. We appreciate that if you could kindly update the score in your review. Thank you!

---

### Official Review · Reviewer_2Umq · 2022-07-11

**Rating:** 6
**Confidence:** 3
**Soundness:** 3 good
**Presentation:** 2 fair
**Contribution:** 3 good

**Summary:**

This paper considered the off-policy evaluation with the utility measured by the output of a downstream decision-making problem, instead of the classical average of individual causal outcomes across a population. The new target was further referred to as policy-dependent linear optimization responses. The authors constructed an unbiased value estimator based on a perturbation method, which addressed the fundamental challenge of an additional optimization bias. Theoretical results such as asymptotic variance properties for a set of adjusted plug-in estimators were derived, followed by extensive numerical simulations.

**Questions:**

Please consider answering the questions mentioned in the Weaknesses above.

**Limitations:**

I didn't find any limitations discussed. Please correct me if I am missing anything.

**Strengths And Weaknesses:**

**Strengths**

- The authors tackled the off-policy evaluation problem from an interesting and novel angle by handling the policy-dependent linear optimization responses.
- Related literature is well discussed and noted.
- The theoretical analyses are sound with promising simulation results.
- This paper is well written and organized.


**Weaknesses**

- The authors may consider improving the presentation of important concepts. For example, are these concepts, *policy-dependent response*, *policy-dependent optimization response*, *downstream
policy-dependent response*, and *downstream decision response* the same or different? An organized introduction of these terms and consistent terminologies may be helpful. Though many examples the authors provided are quite helpful for understanding the policy-dependent response, I would expect an earlier high-level introduction of these important concepts in Section 1 and more detailed explanations of a decision problem $x$ on $m$ units in Section 2.

- Though the authors show that Proposition 4.3 implies asymptotic unbiasedness of the proposed debiased estimator, I wonder if there is any possibility of getting the convergence rate of the policy evaluator as has been done by other recent OPE methods [Athey et al., 2017, Kitagawa and Tetenov, 2018, Kallus and Zhou, 2018b]?


Minor:

- Please define and detail terms before using their abbreviation, such as augmented inverse probability weighted for AIPW in line 86, conditional average treatment effect for CATE in line 216, and GMM in line 246?
- Please consider introducing the notation before using it, such as $A$ in Equation (1).

---

> ### Author Response · Authors · 2022-08-02
> **Author response to reviewer 2Umq**
>
> Dear reviewer,
>
> Thank you for your thoughtful review! We hope our response answers your questions.
>
> # presentation of the important concepts
>
> Thank you for the suggestion, we will add a paragraph end of Section 1 to explicitly define the terminology we use. In particular, the terms you pointed out (e.g. policy-dependent response..etc) are indeed referring to the same concept.
>
> We will also add a detailed explanation of “the decision problem $x$ on $m$ units” at the beginning of Section 2. In particular, $m$ represents the dimension of the downstream decision problem, for example in the bipartite matching example (Example 2.3), $m$ is the total number of edges to select from; $x_i=1, i \in \{1\cdots m\}$ means the $i-th$ edge is selected in the matching.
>
> # convergence rate
>
> Obtaining a convergence rate of the policy evaluation with respect to causal effect estimation is simply a byproduct of MSE convergence of (conditional) causal effect estimation and Cauchy-Schwarz inequality applied to the optimization variable. It is quite different from policy learning guarantees so we have omitted the comparison. And, the result is not novel, it applies generically to optimization with estimated coefficients: see [Nam-Ho Nguyen and Fatma Kilinc-Karzan; Risk guarantees for end-to-end prediction and optimization processes] for sharper results or [Hu, Mao, Kallus; Fast Rates for Contextual Linear Optimization for the cauchy schwarz argument]. Because there is no difference in this case to the results for contextual optimization we omitted it from the main text. We will clarify this further in the paper.
>
>
> # presentation of abbreviation and notations
>
> Thank you for spotting it! We will make sure to define all the abbreviations and notations first time using them.

---

### Official Review · Reviewer_k7Xd · 2022-07-14

**Rating:** 7
**Confidence:** 3
**Soundness:** 4 excellent
**Presentation:** 3 good
**Contribution:** 4 excellent

**Summary:**

This paper studied a new framework for off-policy evaluation, where there is a personalized treatment policy on individual-level outcomes and a policy-dependent optimization response. Unlike previous work focus on the average outcome of individuals, this new framework concerns the output of the downstream decision making or planning problems. Under the new framework, the authors  propose a general algorithm that addresses the challenge of optimization bias for policy evaluation. Empirical synthetic results show the effectiveness of the proposed method under this new framework.

**Questions:**

- The authors mainly focus on performing synthetic dataset experiments to validate the estimation properties. Do the authors think it is possible to conduct more realistic experiments? Though it might be challenging, I feel it would be a strong argument to support your claim that the new framework might be more important in practice.

- Can you explain the computational cost of the proposed method? What if we have really large dataset?

**Limitations:**

There is no potential negative societal impact of this work.

**Strengths And Weaknesses:**

- **Originality**.
The authors proposed to study a new settings / framework that focus on the output of downstream decision-making problems, which is quite interesting and difference from prior works. As far as I know, this new framework has not been studied in previous off-policy evaluation / optimization works.

- **Quality**.
This paper is technically sound. The authors discussed the estimation and optimization bias under this new framework, and shows how to adjust the optimization bias based on a perturbation method. The authors also validate their method on synthetic data to support their claim.

- **Clarity**.
The paper is well organized and well-written. The authors first introduce the problem in Section 2 and the challenge of optimization bias in the proposed setting in Section 3, later the authors discussed how to adjust the optimization bias and conduct empirical experiments in Section 5. The whole story line is well organized.
A small suggestion is that the authors may add some figures to explain the example 2.3

- **Significance**.
This paper studied a new framework for off-policy evaluation. Since the focus of this work is quite different from previous where people concerns more about the average outcome, potentially this work may bring many new perspectives for causal policy evaluation and optimizations.

---

> ### Author Response · Authors · 2022-08-02
> **Author response to reviewer k7Xd**
>
> Dear reviewer,
>
> Thank you for your thoughtful review! We hope our response answers your questions.
>
> ## computational costs
>
> All of our evaluations were run on a 2.3 GHz 8-Core Intel Core i9 CPU.
> For causal effect estimation, we used 2000 random samples for all methods and the total runtime with 20 draws of samples was less than 2 seconds. For policy evaluation, the average runtime with 500-2000 training samples, 20 bootstrap replications was within 12-15 seconds (longer time due to bootstrap). For policy optimization we used training datasets with size 1000 where one sub-gradient descent iteration took around 9 second, and as the results showed all methods typically converge within 50 iterations.
>
> Though improving the computation efficiency was not a main focus of this paper, we believe that the method, especially the bootstrap procedure in the perturbation algorithm is easy to parallelize and run with multi-processing (e.g. Python thread-based pool), which can improve the runtime significantly.
>
> We will include the runtime details in the appendix.
>
>
> ## real-data experiments
>
> We are not aware of real data that is suitable for both policy evaluation and downstream optimization (potentially due to privacy reasons).  However, we tried to illustrate the advantages of the proposed approach over different benchmark estimators through various stress-tested scenarios, and conducted more comparisons to other nonparametric estimators (e.g. causal forests [Wager and Athey, 2018]).

---

> > ### Comment · Reviewer_k7Xd · 2022-08-09
> > **Thanks for your clarification**
> >
> > Thank the authors for the response. I have no further questions or concerns.

---

### Official Review · Reviewer_2odh · 2022-07-18

**Rating:** 7
**Confidence:** 3
**Soundness:** 3 good
**Presentation:** 3 good
**Contribution:** 3 good

**Summary:**

The paper considers evaluation and optimization of a policy that selects a binary treatment for each individual, who then interact via an optimization problem. The challenge, as it usually is with policy evaluation, is imputing the counterfactual outcomes that depend on the treatments assigned by the policy, which may not be the ones observed in the data. They consider the case where the counterfactuals are the coefficients of the optimization objective (or, thanks to duality, the constraints), which may be performed implicitly by a market, or explicitly by an operations system, after observing the outcomes induced by the policy. This differs from other perspectives on this problem, which focus on estimating the per-individual treatment effect after optimization, introducing serious challenges in SUTVA violations. The structure of the optimization problem and the policy evaluation / learning objective circumvents these challenges.

**Questions:**

The main text says that Appendix B discusses why AIPW cannot be used, but I couldn't find any such discussion. Can you please point me to that?

The paper mentions that Shi et al. [2019] uses the GRDR method for CATE estimation, but the referenced paper seems to focus on ATE estimation. Can you please clarify?

The GRDR method seems potentially challenging for fitting with certain black box ML methods, but it looks aesthetically similar to the TMLE, with $\hat\mu$ fit first, and then $\epsilon_1, \epsilon_0$ fit afterwards. Would this have the same nice properties as the GRDR method that the AIPW lacks? Is it important that $\hat\mu$ is fit at the same time?

If the plug-in estimate with GRDR is equivalent to the AIPW (as quoted from Bang and Robins), and the AIPW cannot directly be used with the downstream optimization problem, why can the GRDR plug-in be used? They don't look quite equivalent to me, but I can't tell what the key property is that GRDR has and AIPW lacks.

**Limitations:**

Methodology for studying the potential implications of automated policies in economic and other social settings is the primary topic of the paper, and is discussed. Negative impact is implicitly discussed via discussion of the biases from using this methodology (and attempts to mitigate it), and via discussion of when post-policy optimization affects the potential individual improvements purportedly created by the policy.

**Strengths And Weaknesses:**

The problem of policy evaluation is a critically important one to the field of machine learning today, and this paper adds a valuable, original, perspective on how to think about policy evaluation and its relationship to treatment effects when individuals interact in a market or other system. The paper gives a convincing demonstration of the significance of the optimization problem structure that they study by giving examples of real, practical problems that can be formulated in their framework.

The issue with this paper is the clarity / soundness of the statistical model for the optimization problem. In Assumption 2.1, the coefficients of the optimization problem depends on the potential outcomes, $c_i(\pi)$. However, in Section 3, the problem changes and the coefficients are (abusing notation to emphasize the similarity) $\mu_\pi(w_i)$. Normally in policy evaluation or causal inference, this difference in irrelevant due to the tower law and the fact that $\mathbb{E}[ c_i(Z_i) \mid w_i, Z_i ] = \mu_{Z_i}(w_i)$. However, in this case, there is no expectation that allows one to use this property of the tower law. So, the objective has simply changed, and it's unclear if the new class of optimization problems is a reasonable / practically significant one to pursue.

One might hope, based on the aesthetics, that Assumption 2.2 wouldn't have this issue, given that the objective has an expectation that would allow the tower property to be used. However, I had trouble conceptually parsing Assumption 2.2, because I did not understand what type of mathematical object $x$ is. At first glance, it seems like a scalar. However, the description says that we should think of this expectation as the limit as we take the number of objects to infinite. So, it seems like x is now a mapping from the sample space of the probability measure for the expectation to $\mathbb{R}$. In this case, $x$ would not be measurable with respect to $w_i$ so that the desired tower property can apply. Additionally, I would wonder about measurability issues, although these are probably resolvable under reasonable assumptions and not a primary concern, just and indication of subtleties that can trip even the careful mathematician up.

---

> ### Author Response · Authors · 2022-08-02
> **Author response to reviewer 2odh**
>
> Dear reviewer,
>
> Thank you for your thoughtful review! We hope our response answers your questions.
>
> ## ..statistical model for the optimization problem..
> We’d like to clear up potential misunderstandings regarding concerns about “clarity / soundness of the statistical model for the optimization problem”. We adopt conventions both from causal inference with potential outcomes and policy evaluation/learning; and contextual/data-driven optimization. Since we are working with standard frameworks in stochastic optimization, we omit the standard measurability framework within which we work. We will include textbook references to Shapiro, Dentcheva, Ruszcynski (2009 edition, pg 156 for discussion of measurability, sec. 5.1.2. for discussion of optimization bias) to further clarify. (The contextual extension to this framework is to have the random vector $\xi$ be distributed according to the conditional law of $\xi|W$.) We will make this clearer.
>
> ### ...in Section 3, the problem changes and the coefficients are (abusing notation to emphasize the similarity) $\mu_\pi(W_i)$ … there is no expectation that allows one to use this property of the tower law...
>
> We assume this refers to line 175; we believe this confusion arises because we are working in a contextual linear optimization framework where decisions $x$ are measurable with respect to observed covariates $W$, but do not observe the random realizations of costs at the time of making decisions. In eq(1) we redefine c_i(\pi) = \E[\mu_\pi | W_i] but this was not very explicit. Given the use of c(\pi) in the previous section we agree this notation is too compact. For consistency we will change eq(1) to be E[\mu_\pi | W_i] (this improves alignment/comparison both with assumption 2.2 and line 186).
>
> This class of optimization problems — optimizing (conditional) expectations of random costs which leads to optimization bias — is indeed the practically significant formulation in optimization under uncertainty. This subtlety was detailed from *line 171-190*. We will make sure to expand on it and make it clearer in the paper. We will inline discussion of the reasonable assumptions (i.e. finiteness of random potential outcomes).
>
> This subtlety was detailed from *line 171-190*. We will make sure to expand on it and make it clearer in the paper.
>
>
> ### Assumption 2.2
>
>  As mentioned, assumption 2.1 is the main focus of the paper because assumption 2.2 is more complicated. We made this distinction to be absolutely clear. To help clarify we will write out the expectation objective to be E[c(\pi) x] = E[ \E[c(\pi)|w] x(w) f(w) dw]; x as an optimization variable is assumed to be in a Banach space and is a function of w. The tower property still applies but it is now an infinite-dimensional linear (contextual) program (hence imposes additional requirements on strong duality). This still remains a fairly standard stochastic optimization problem.
>
> As detailed in line 107-117, our policy $\pi$ maps from covariates to a (probability of) treatment. The outcome $c(\pi)$ has two sources of randomness: the covariate $w_i$, and the random treatment that is assigned by the policy. Assumption 2.2 takes an expectation over these two sources of randomenss: the outer expectation is taken over the randomness of the policy, and the inner expectation is taken over the randomness of $w_i$.
>
> We will make sure to make this clear in the paper.

---

> > ### Comment · Reviewer_2odh · 2022-08-08
> > **Thanks for clarification**
> >
> > Hi,
> >
> > Thanks for your clarification on the optimization bias. When I re-read the section, it was clear that this was, in fact, discussed. I believe that the confusion was with the section that follows in lines 181-190 (in particular, 187-188). In these lines, the emphasis seems to be on taking the expectation first and then optimizing, versus the other way around. In particular, none of the quantities written out here are about $c(\pi)$ versus $\mu_\pi(w)$.
> >
> > In reading it a second time, it seems like perhaps 181-190 is about a second form of optimization bias, different from $\min_x \sum_{i} c_i(\pi) x_i$ versus $\min_x \sum_{i} \mu_\pi(w_i) x_i$ which is previously discussed. Or perhaps there is some subtlety here making them equivalent that I am misunderstanding. Either way, more clarity here and caution around notation would help make this easier to follow.
> >
> > In short, it seems like there are many pieces here:
> >   1) $\min_x \sum_{i} c_i(\pi) x_i$ versus $\min_x \sum_{i} \mu_\pi(w_i) x_i$, which is mentioned as the optimization bias in 170-180
> >   2) $\min_x \sum_{i} \mu_\pi(w_i) x_i$ versus $\min_x \sum_{i} \hat{\mu}_\pi(w_i) x_i$ which is mentioned as the estimation bias in 170-180
> >   3) $\mathbb{E}[\min_x \sum_{i} c_i(\pi) x_i]$ versus $\min_x \mathbb{E}[\sum_{i} c_i(\pi) x_i]$, or something like this, discussed in 181-190
> > The last piece is a bit unclear to me, which would be helpful to understand the overall point. Given the comment made about $x$ really being measurable with respect to $w$ and not $c$, I would encourage making this clear in the model directly. This would help clarify the notation in the above discussion, and clarify how the examples given in the paper fit with this assumption.
> >
> > Regarding Assumption 2.2, I would definitely appreciate a more verbose but explicit definition of the optimization problem, as mentioned. I agree that it does not seem to be the main model studied in the paper, and therefore isn't as important, but it helps clarify some of the confusion around what $x$ and $c(t)$ throughout the paper can depend on.
> >
> > Finally, I don't think it's necessary to spend more time / words on the measurability questions. To me, they are an indication of something unclear going on with the definition of the optimization problem, rather than a concern about formality.
> >
> > In light of this clarification and the authors agreeing to make some improvements, I'm willing to increase my score. I do want to note that the authors have the ability to update the pdf, and I would feel much more comfortable accepting the paper after seeing the updated clarifications. However, this has been a confusing aspect of changing to OpenReview and the standard for this conference in the past has been to expect authors to deliver on the clarifications promised in rebuttals, so I will simply increase the score now.

---

> ### Author Response · Authors · 2022-08-02
> **response to 2odh questions**
>
> # Responses to questions
>
> ### why AIPW cannot be used
>
> We will fix this reference to the appendix to Appendix C, the referenced discussion is in lines 669-678.
>
> ### AIPW, GRDR
>
> Thanks for pointing out a potential confusion with the presentation. In that section, we had presented the methods with their original motivations (and so we presented some of the original motivation for GRDR and targeted regularization). As you point out, the original papers are motivated specifically for ATE estimation which can be confusing: we introduce GRDR as a possibility because unlike direct computation of an estimator like AIPW, the method of GRDR will lead to a prediction function which can be queried on out-of-sample covariates.  Note that in this case, pseudo-outcomes that marginalize to ATE estimate could also be used for conditional treatment effect (CATE) estimation by regressing again on the output of GRDR, following the pseudooutcome regression approach used in Kennedy 2020 and Semenova and Chernozhukov 2018 for best linear projections of CATE. That is, regressing \mu_GRDR(X) again on X will give a valid treatment outcome estimator.
>
> To remedy this point of potential confusion, we will include a brief remark about GRDR and mention explicitly that we explore its applicability as another prediction function (in contrast to its primary motivations). Appendix lines 687-691 discuss these differences, but we will adjust the main text and move this motivation for GRDR that is not applicable in the main regime we study to the appendix.
>
> ### ... similarities between GRDR and TMLE ...
>
> Astute observation, indeed GRDR was studied recently in Shi et al under the name "targeted regularization", motivated as an algorithmic simplification of TMLE. This would have similar properties as GRDR that AIPW lacks because TMLE also results in a prediction function $\mu_t(W)$ that can be queried on out-of-sample data. It's not necessary that $\hat \mu$ is fit at the same time.
>
> ### ... why can the GRDR plug-in be used? ...
>
> See discussion in "AIPW, GRDR". We agree it can be confusing to present the initial motivations for GRDR in a setting where AIPW cannot be used. We will clarify that we present it as an example of a causal-aware estimation method that results in a prediction function $\mu_t(W)$ that can be queried on out-of-sample data; for example we will also present "metalearners" from Sekhon et al. such as the S-learner, etc. as other examples of causal-aware estimation with this property.

---

### Comment · Area_Chair_RqNu · 2022-08-06
**Please make sure the authors' response has been read**

Dear Reviewers,

Thanks for providing the reviews. The discussion stage will end in next Tuesday. Please check the authors' response and feel free to discuss with authors.

Best, AC

---

### Meta-Review · Area_Chair_RqNu · 2022-08-30

**Recommendation:** Accept
**Confidence:** Certain

**Metareview:**


In the paper, the authors considered a new formulation for the off-policy evaluation and optimiation, in which the utility measured by the output of a downstream decision-making problem, therefore, the policy value is dependent on individual reponse. This difference induces additional optimization bias in the estimand and requires new technique to handle it. The authors constructed estimators by combining perturbation method with IPW estimators, and provided the theoretical guarantees and empirical study.

Most of the reviewers provide positive feedback on this paper. It will be great if the authors can take the authors suggestions into account to improve the papers:

- Although as the authors explained the finite sample analysis is not new, for completeness, I think it will be great if this can be added into the paper.

- As reviewer suggeseted, the method is motivated well from practical problems, however, has not justified with real-world applications. The paper will be significantly improved if this part can be added.


Finally, there is naturally technique, interchangeability principle [1, 2], can be used to bypass the optimization bias, besides the perturbation method, which should be discussed and compared.

> [1] Dai, Bo, Niao He, Yunpeng Pan, Byron Boots, and Le Song. "Learning from conditional distributions via dual embeddings." In Artificial Intelligence and Statistics, pp. 1458-1467. PMLR, 2017.\
[2] Shapiro, A., and Dentcheva, D. (2014). Lectures on stochastic programming: modeling and theory. SIAM (Vol. 16).

**Award:**

No

---

### Decision · Program_Chairs · 2022-09-14

Accept